# Vascular imbalance in polycystic ovary syndrome: Insights into endothelial dysfunction

**Tuğba Raika Kıran** [ORCID][1]*, **Mehmet Erdem** [ORCID][1], **Engin Yıldırım**[2], **Feyza İnceoğlu**[3]

**1** Department of Medical Biochemistry, Faculty of Medicine, Malatya Turgut Özal University, Malatya, Turkey, **2** Department of Obstetrics and Gynecology, Faculty of Medicine, Malatya Turgut Özal University, Malatya, Turkey, **3** Department of Biostatistics, Faculty of Medicine, Malatya Turgut Özal University, Malatya, Turkey,

* raika.kiran@ozal.edu.tr

## Abstract

Polycystic ovary syndrome (PCOS) is a multifactorial endocrine disorder associated with vascular dysfunction and increased cardiovascular risk. This study aims to investigate the dysregulation of vascular tone in PCOS, focusing on the imbalance between vasodilators (nitric oxide [NO] and apelin) and vasoconstrictors (noradrenaline and reduced prostacyclin). By examining these factors, the study seeks to elucidate their contribution to endothelial dysfunction and cardiovascular complications in PCOS patients. Forty-four patients diagnosed with PCOS according to the 2003 Rotterdam Criteria, along with 44 healthy controls, were included in the study. Ultrasound evaluations were performed on all volunteers. Serum NO, apelin, noradrenaline, and prostacyclin levels were measured using commercial enzyme-linked immunosorbent assay (ELISA) kits. Additionally, routine biochemical, hormonal, and glycated hemoglobin analyses were conducted on all samples. There was no statistically significant difference between the PCOS and control groups in terms of marital status, age, and body mass index (BMI) ($p > 0.05$). Compared to the control group ($83.85 \pm 22.65$ μmol/L, $190.88 \pm 16.44$ ng/L, and $24.63 \pm 4.59$ ng/L, respectively), NO, apelin, and noradrenaline concentrations were significantly higher in patients with PCOS ($104.35 \pm 44.96$ μmol/L, $379.57 \pm 40.11$ ng/L, and $27.48 \pm 5.36$ ng/L, respectively) ($p < 0.01$). In contrast, prostacyclin concentrations were significantly lower in patients with PCOS ($5.85 \pm 1.28$ ng/L) compared to the control group ($6.78 \pm 1.99$ ng/L) ($p = 0.011$). Additionally, a statistically significant difference was found between the PCOS and control groups in FSH, LH, testosterone, SHBG, glucose, and HDL levels ($p < 0.05$). The disrupted balance between vasodilation and vasoconstriction in PCOS, driven by altered levels of NO, apelin, noradrenaline, and prostacyclin, contributes to endothelial dysfunction and increased cardiovascular risk. These molecular disturbances underline the need for targeted therapeutic strategies aimed at restoring vascular homeostasis in PCOS patients.

**Data availability statement:** All relevant data are within the manuscript and its Supporting Information files.

**Funding:** The author(s) received no specific funding for this work.

**Competing interests:** The authors have declared that no competing interests exist.

## Introduction

Polycystic ovary syndrome (PCOS) is a prevalent endocrine and metabolic disorder among women of reproductive age, characterized by hyperandrogenism, menstrual irregularities, and polycystic ovarian morphology [1]. The condition affects approximately 8% to 13% of women globally, with significant implications for metabolic health, reproductive function, and psychological well-being [2]. Although the exact etiology of PCOS remains unknown, it is believed to result from a complex interplay of genetic predisposition, environmental determinants, and lifestyle factors [3–5]. PCOS is associated with metabolic complications such as insulin resistance, obesity, type 2 diabetes, and an increased risk of cardiovascular diseases [6–8]. The widely adopted Rotterdam Criteria for PCOS diagnosis encompass the presence of hyperandrogenism, ovulatory dysfunction, and polycystic ovarian morphology confirmed by ultrasound [9]. Management of PCOS involves lifestyle modifications such as diet and exercise, along with pharmacological interventions. Common treatment options include oral contraceptives, insulin sensitizers, and anti-androgen medications. Current research is focused on understanding the long-term effects of PCOS and developing personalized therapeutic approaches [10,11].

Nitric oxide (NO) is a critical signaling molecule involved in numerous biological functions, including vasodilation, neurotransmission, and immune response. NO is implicated in the regulation of ovarian function and the modulation of insulin sensitivity [12]. In recent decades, considerable focus has been placed on the involvement of NO in regulating key reproductive processes, such as follicular development, hormone production, granulosa cell apoptosis in atretic follicles, and changes in oocyte maturation potential. Additionally, NO is thought to influence the activity of cyclooxygenases (COX-1, COX-2) and prostaglandin synthesis essential for ovulation. The ovaries have the ability to produce NO, which may play a crucial role in ovarian steroidogenesis, ovulation, and corpus luteum regression [13,14].

Originally identified from bovine stomach extracts as the natural ligand for the orphan APJ receptor, a member of the G-protein-coupled receptor family, apelin is expressed in ovarian tissues, such as granulosa and theca cells, and is considered to play a significant role in regulating ovarian function [15,16]. Apelin exhibits both vasodilatory and vasoconstrictive effects, depending on the physiological context, as its interaction with APJ receptors varies between vascular smooth muscle cells (VSMCs) and endothelial cells, while it generally promotes NO-mediated vasodilation [17], it can induce vasoconstriction in the absence of endothelial influence [18].

Noradrenaline is a fundamental neurotransmitter and hormone that regulates various physiological processes, particularly within the cardiovascular system. It plays a pivotal role in modulating vascular tone by mediating vasoconstriction. The interplay between noradrenaline and other signaling molecules, such as NO and prostacyclin, is critical for maintaining vascular homeostasis and adapting to physiological demands [19,20]. The regulation of vascular tone by noradrenaline is influenced by the balance between vasoconstrictors and vasodilators [20]. Prostacyclin is a potent vasodilator and inhibitor of platelet aggregation, playing a crucial role in maintaining

vascular homeostasis. Its synthesis occurs primarily in endothelial cells, where it is derived from arachidonic acid through the cyclooxygenase (COX) pathway [21]. Changes in the morphology of periovarian adipose tissue, which plays a crucial role in ovarian function by secreting bioactive molecules such as adipokines, cytokines, NO, and prostacyclin to support vascular homeostasis and regulate vascular tension, have been observed in conditions such as menopause and PCOS [22].

Dysregulation of these vasoactive molecules has also been implicated in various reproductive and vascular disorders. For instance, altered NO signaling is associated with abnormal angiogenesis in endometriosis and ovarian hyperstimulation syndrome [14,23]. Elevated apelin levels have been reported in obese and PCOS-related follicular arrest [24], while increased noradrenaline activity contributes to sympathetic overactivity and disrupted ovarian function in PCOS [25]. Given the well-established vasodilatory and anti-thrombotic functions of prostacyclin, and its protective role in endothelial homeostasis as observed in pulmonary hypertension [26], it is plausible that reduced prostacyclin bioavailability may also contribute to endothelial dysfunction in PCOS, which is characterized by impaired endothelium-dependent vasodilation and insulin resistance [27]. These findings suggest that these molecules not only support physiological processes but also participate in pathophysiological mechanisms relevant to ovarian and metabolic diseases. These four biomarkers were selected due to their complementary roles in vascular tone regulation, including endothelial relaxation (NO and prostacyclin), vasoconstriction (noradrenaline), and angiogenic signaling (apelin), all of which are known to be altered in PCOS. Given that PCOS is characterized by endothelial dysfunction, chronic low-grade inflammation, and insulin resistance, simultaneous evaluation of these markers allows for a more integrative understanding of the pathophysiological disturbances underlying vascular dysregulation in affected individuals. Although the physiological roles of NO, apelin, noradrenaline, and prostacyclin have been individually investigated, no study to date has systematically examined their combined contribution to vascular tone imbalance in PCOS. Furthermore, their collective diagnostic value in distinguishing PCOS patients from healthy individuals remains largely unexplored. This study addresses this gap by providing an integrative evaluation of these markers, thereby offering a more comprehensive approach to understanding endothelial dysfunction in PCOS.

We hypothesize that the characteristic hormonal and metabolic alterations in PCOS -including hyperandrogenism, insulin resistance, and chronic inflammation- may disrupt the balance between these vasodilatory and vasoconstrictive factors, contributing to endothelial dysfunction and cardiovascular risk.

This study aims to comprehensively explore the relationship between serum NO, apelin, noradrenaline, and prostacyclin levels with vascular tone regulation and dysfunction in women with PCOS and to delineate their potential contributions to the pathophysiological mechanisms underlying PCOS.

## Materials and methods

### Study design, participants, exclusion criteria, and ethical approval

This study was conducted between August 2022 and November 2022 at the Department of Obstetrics and Gynecology Faculty of Medicine Malatya Turgut Özal University, Malatya, Turkey. The patient and control groups in the study were composed of volunteers who applied to Faculty of Medicine Malatya Turgut Özal University. At the beginning of the study, the individuals' medical histories were taken. Demographic and anthropometric data, including marital status, age, weight, height, and body mass index (BMI), were recorded and ultrasound evaluation (Samsung RS85 Prestige, Gangwon-do, Republic of Korea) was performed. Forty-four individuals aged 18–40 years diagnosed with PCOS according to Rotterdam Criteria were included in the study [28]. According to the 2003 Rotterdam Criteria, a diagnosis of PCOS is confirmed when at least two of the following conditions are present: (1) oligomenorrhea or amenorrhea, (2) clinical or biochemical signs of hyperandrogenemia, and (3) polycystic ovarian morphology detected through ultrasonography [29]. Individuals who were pregnant or breastfeeding, or those with irregular menstrual cycles, androgen excess conditions (such as Cushing's

syndrome), gestational diabetes mellitus, arterial disease, hypertension, congestive heart failure, chronic liver or kidney failure, type 1 or type 2 diabetes, and hyperlipidemia were excluded from the study. Forty-four individuals of the same ethnic origin, aged between 18 and 40 years, with similar demographic characteristics, who met the exclusion criteria, had no diseases, had regular menstrual cycles, and had normal ovarian morphology on ultrasonography, were included in the control group. To minimize hormonal variability, blood samples from cycling control participants were collected during the early follicular phase (days 2–5) of the menstrual cycle.

Written informed consent was obtained from all volunteers in the study. The study was conducted according to the Declaration of Helsinki. The study was approved by the Ethics Committee of Malatya Turgut Özal University, Turkey (Number: 2022/35, Date: July 28, 2022). The collected data were accessed between 01.12.2022 and 05.01.2023 for research purposes.

## Blood collection and biochemical analysis

Early in the morning, after an overnight fast, blood samples were collected by a single specialist phlebotomist, from the brachial veins of all volunteers into two gel separator tubes (serum collection) and one $K_2$EDTA tube. After the collection process, all tubes were properly transported to the biochemistry laboratory for routine analysis. Glycated hemoglobin (HbA1c) measurements in $K_2$EDTA tubes were performed using an automatic glycohemoglobin analyzer (Arkray, Adams A1c HA-8180V, Japan). The serum collection tubes were left for 20–30 minutes to allow clotting. Once coagulation was complete, the serum collection tubes were centrifuged at 1800 g for 10 minutes. First serum collection tube was used for biochemical analysis (Glucose, triglyceride, total cholesterol, low-density lipoprotein [LDL], and high-density lipoprotein [HDL]) and hormone analysis (Follicle-stimulating hormone [FSH], luteinizing hormone [LH], estradiol [E2], thyroid-stimulating hormone [TSH], prolactin, human chorionic gonadotropin [β-hCG], testosterone, sex hormone-binding globulin [SHBG], dehydroepiandrosterone sulfate [$DHESO_4$], and insulin). The analyses were conducted using a biochemistry analyzer (Abbott Architect c16000, Illinois, USA) and a hormone analyzer (Roche Diagnostics Cobas E601, Tokyo, Japan), respectively. Serum samples in the second serum collection tube (obtained at the end of centrifugation) were transferred to 1.5 mL micro-volume tubes. These serum samples were placed in a −80°C deep freezer for NO, apelin, noradrenaline, and prostacyclin analyses and stored there until the analyses were conducted.

On the day of analysis, serum NO, apelin, noradrenaline, and prostacyclin (Bioassay Technology Laboratory, Cat. No: E1510Hu, E2014Hu, EA0069Hu, and EA0019Hu, China respectively) levels were measured using human-specific enzyme-linked immunosorbent assay (ELISA) kits, following the manufacturer's instructions.

## Statistical analysis

The data collected for the study were analyzed using SPSS (Statistical Program for Social Sciences) version 25. The normality of the data was assessed using the Kolmogorov-Smirnov test. Descriptive data are presented as mean, standard deviation, frequency, and percentage values. The significance level (p) for comparison tests was set at 0.05. Since the data followed a normal distribution (p > 0.05), parametric test methods were used for further analysis. Comparisons between independent two groups were made using the t-test, as the normality assumption was met. To determine the cutoff point for a measurement value, ROC analysis was performed and indices were calculated. Binary logistic regression models were established in which the groups were the dependent variables and the NO, apelin, noradrenaline, and prostacyclin values were the independent variables. The Hosmer-Lemeshow statistic was used to test the model's goodness of fit in the binary logistic regression analysis. The sample size for this study was calculated using the G*Power 3.1 program. According to the results, a minimum of 84 participants (42 per group) was required, with an effect size of 0.73, a margin of error of 0.05, a confidence level of 95%, and a statistical power of 0.95 [30].

## Results

### Basic demographic and anthropometric data

Forty-four female patients with PCOS and 44 female healthy controls were included in the study. No significant difference was observed between the PCOS and control groups in terms of marital status (p = 0.243) (Table 1). The mean ages of PCOS patients and control subjects were 29.23 ± 4.2 and 29.5 ± 6.82, respectively. No significant difference was observed between the PCOS and control groups in terms of age (p = 0.822) (Table 1). The mean BMI value of PCOS patients and control subjects were 22.54 ± 2.06 and 21.91 ± 2.01, respectively. No significant difference was observed between the PCOS and control groups in terms of BMI (p = 0.151) (Table 1).

### Evaluation of clinical laboratory parameters

Within the study, no statistically significant differences were observed between the PCOS and control groups in terms of estradiol, TSH, prolactin, β-hCG, DHESO$_4$, insulin, HOMA-IR, HbA1c, triglyceride, total cholesterol, and LDL levels (p > 0.05) (Table 2). However, FSH, LH, testosterone, SHBG, glucose, and HDL levels were found to be statistically significant higher in the PCOS group compared to the control group (p < 0.05) (Table 2).

### Evaluation of NO, apelin, noradrenaline, and prostacyclin levels

The mean NO levels of PCOS patients and control subjects were 104.35 ± 44.96 and 83.85 ± 22.65, respectively. NO levels were found to be significantly higher in the PCOS group compared to the control group (p = 0.008) (Table 3). The mean apelin levels of PCOS patients and control subjects were 379.57 ± 40.11 and 190.88 ± 16.44, respectively. Apelin levels were found to be significantly higher in the PCOS group compared to the control group (p = 0.0001) (Table 3). The mean noradrenaline levels of PCOS patients and control subjects were 27.48 ± 5.36 and 24.63 ± 4.59, respectively. Noradrenaline levels were found to be significantly higher in the PCOS group compared to the control group (p = 0.009) (Table 3). The mean prostacyclin levels of PCOS patients and control subjects were 5.85 ± 1.28 and 6.78 ± 1.99, respectively. Prostacyclin levels were found to be significantly lower in the PCOS group compared to the control group (p = 0.011) (Table 3).

### Logistic regression analysis

According to binary logistic regression models, the model established with NO, apelin, noradrenaline, and prostacyclin variables was found to be statistically sufficient for predicting the data groups ($\chi^2$ = 12.931, df = 8, p = 0.114; $\chi^2$ = 8.492, df = 7, p = 0.291; $\chi^2$ = 14.079, df = 8, p = 0.080; $\chi^2$ = 4.76, df = 8, p = 0.783, respectively). It was found that the measurement

**Table 1. Comparison of the basic demographic and anthropometric data of patients with PCOS and healthy controls.**

| Variable | Group | n / % | Groups | | Total | $\chi^2$ | p |
|---|---|---|---|---|---|---|---|
| | | | PCOS | Control | | | |
| Marital Status | Married | n | 16 (36.36%) | 10 (22.73%) | 26 (29.55%) | 1.365 | 0.243 |
| | Single | n | 28 (63.64%) | 34 (77.27%) | 62 (70.45%) | | |
| Variable | | | PCOS | Control | | t value | p value |
| | | | Mean ± SD | Mean ± SD | | | |
| Age | | | 29.23 ± 4.2 | 29.5 ± 6.82 | | -0.226 | 0.822 |
| BMI | | | 22.54 ± 2.06 | 21.91 ± 2.01 | | 1.449 | 0.151 |

n, frequencies; %, percent; PCOS, polycystic ovary syndrome; SD, standard deviation; BMI, body mass index; p < 0.05, there is a statistical difference between the groups.

**Table 2. Comparison of the hormonal and metabolic parameter data between patients with PCOS and healthy controls.**

| Variable | PCOS | Control | t value | p value |
|---|---|---|---|---|
| | Mean±SD | Mean±SD | | |
| FSH (mIU/mL) | 5.96±1.51 | 8.05±2.52 | −4.694 | **0.001** |
| LH (mIU/mL) | 8.66±5.08 | 6.16±2.05 | 3.018 | **0.003** |
| Estradiol (pg/mL) | 50.13±28.13 | 56.73±29.28 | −1.078 | 0.284 |
| TSH (mU/L) | 1.64±0.66 | 1.74±0.95 | −0.582 | 0.562 |
| Prolactin (ng/mL) | 15.53±7.25 | 13.58±6.64 | 1.313 | 0.193 |
| β-hCG (mIU/mL) | 0.3±0.61 | 0.2±0.02 | 1.119 | 0.266 |
| Testosterone (ng/dL) | 0.33±0.18 | 0.24±0.1 | 2.808 | **0.006** |
| SHBG (nmol/L) | 46.5±17.75 | 56.94±23.99 | −2.320 | **0.023** |
| DHESO$_4$ (ng/dL) | 201.55±82.59 | 177.32±89.2 | 1.322 | 0.190 |
| Insulin (uU/mL) | 8.61±2.27 | 8.34±2.26 | 0.557 | 0.579 |
| Glucose (mg/dL) | 88.66±6.9 | 91.36±5.05 | −2.097 | **0.039** |
| HOMA-IR | 1.89±0.52 | 1.88±0.51 | 0.079 | 0.937 |
| HbA1c (mmol/L) | 5.34±0.22 | 5.29±0.24 | 0.951 | 0.344 |
| Triglyceride (mg/dL) | 96.93±27.82 | 94.52±31.2 | 0.382 | 0.703 |
| Total cholesterol (mg/dL) | 158.25±24.26 | 152.98±30.65 | 0.895 | 0.373 |
| LDL (mg/dL) | 97.55±16.35 | 91.46±16.41 | 1.744 | 0.085 |
| HDL (mg/dL) | 52.08±7.15 | 48.08±6.8 | 2.692 | **0.009** |

PCOS, polycystic ovary syndrome; SD, standard deviation; t, independent t-test; FSH, follicle-stimulating hormone; LH, luteinizing hormone; TSH, thyroid-stimulating hormone, β-hCG, human chorionic gonadotropin; SHBG, sex hormone-binding globulin; DHESO$_4$, dehydroepiandrosterone sulfate; HOMA-IR, homeostasis model assessment for insulin resistance; HbA1c, glycated hemoglobin; LDL, low-density lipoprotein; HDL, high-density lipoprotein; $p < 0.05$, there is a statistical difference between the groups (Bold values).

**Table 3. Comparison of nitric oxide, apelin, prostacyclin, and noradrenaline levels between patients with PCOS and healthy controls.**

| Variable | PCOS | Control | t value | p value |
|---|---|---|---|---|
| | Mean±SD | Mean±SD | | |
| NO (µmol/L) | 104.35±44.96 | 83.85±22.65 | 2.701 | **0.008** |
| Apelin (ng/L) | 379.57±40.11 | 190.88±16.44 | 28.875 | **0.0001** |
| Noradrenaline (ng/L) | 27.48±5.36 | 24.63±4.59 | 2.674 | **0.009** |
| Prostacyclin (ng/L) | 5.85±1.28 | 6.78±1.99 | −2.606 | **0.011** |

PCOS, polycystic ovary syndrome; NO, nitric oxide; SD, standard deviation; t, independent t-test; $p < 0.05$, there is a statistical difference between the groups (Bold values).

values of NO, apelin, noradrenaline, and prostacyclin had a statistically significant effect in predicting the difference between the PCOS and control groups ($p = 0.012$, $p = 0.0001$, $p = 0.019$, $p = 0.007$, respectively) (Table 4). The difference between the study and control groups is 1.017 times more affected by the NO value (OR = 1.017, 95% CI 1.004–1.031). An increase of 1 unit in the NO value reduces the disease risk by 1.017 times (Table 4). The difference between the study and control groups is 1.044 times more affected by the apelin value (OR = 1.044, 95% CI 1.025–1.064). An increase of 1 unit in the apelin value reduces the disease risk by 1.044 times (Table 4). The difference between the study and control groups is 1.115 times more affected by the noradrenaline value (OR = 1.115, 95% CI 1.018–1.221). An increase of 1 unit in the noradrenaline value reduces the disease risk by 1.115 times (Table 4). The difference between the study and control groups is 0.658 times more affected by the prostacyclin value (OR = 0.658, 95% CI 0.486–0.891). An increase of 1 unit in the apelin value reduces the disease risk by 0.658 times (Table 4).

**Table 4. Logistic regression analysis of nitric oxide, apelin, prostacyclin, and noradrenaline in PCOS patients.**

| Model | Variable | β | SE | W | df | p | Exp (β) (OR) | 95% CI for Exp (β) | |
|---|---|---|---|---|---|---|---|---|---|
| | | | | | | | | Lower Bound | Upper Bound |
| Model 1 | NO | 0.017 | 0.007 | 6.243 | 1 | 0.012 | 1.017 | 1.004 | 1.031 |
| | Constant | −1.573 | 0.660 | 5.683 | 1 | 0.017 | 0.207 | | |
| Model 2 | Apelin | 0.043 | 0.009 | 21.198 | 1 | 0.0001 | 1.044 | 1.025 | 1.064 |
| | Constant | −12.697 | 2.958 | 18.428 | 1 | 0.0001 | 0.001 | | |
| Model 3 | Noradrenaline | 0.109 | 0.046 | 5.511 | 1 | 0.019 | 1.115 | 1.018 | 1.221 |
| | Constant | −2.841 | 1.229 | 5.341 | 1 | 0.021 | 0.058 | | |
| Model 4 | Prostacyclin | −0.418 | 0.155 | 7.304 | 1 | 0.007 | 0.658 | 0.486 | 0.891 |
| | Constant | 2.646 | 0.995 | 7.074 | 1 | 0.008 | 14.098 | | |

NO, nitric oxide; β, parameter estimate; SE, standard error; W, Wald statistic; df, degrees of freedom; Exp (β), odds ratio; 95% CI, confidence interval; $p < 0.05$, there is a statistical difference (Bold values).

## ROC curve analysis

When the cut-off value for NO was set at 44.17 µmol/L, it demonstrated 97.7% sensitivity and 93.2% specificity in detecting PCOS (AUC = 0.536, 95% CI = 0.404–0.667, p = 0.046). When the cut-off value for apelin was set at 161.34 ng/L, it demonstrated 98.7% sensitivity and 95.5% specificity in detecting PCOS (AUC = 0.983, 95% CI = 0.935–0.991, p = 0.001). When the cut-off value for noradrenaline was set at 18.19 ng/L, it demonstrated 93.2% sensitivity and 95.5% specificity in detecting PCOS (AUC = 0.659, 95% CI = 0.542–0.775, p = 0.010). When the cut-off value for prostacyclin was set at 4.23 ng/L, it demonstrated 88.6% sensitivity and 95.5% specificity in detecting PCOS (AUC = 0.351, 95% CI = 0.236–0,466, p = 0.016) (Fig 1).

## Discussion

This study highlights a significant dysregulation in vascular tone among women with PCOS, characterized by an imbalance between vasodilatory (NO and apelin) and vasoconstrictive (noradrenaline and reduced prostacyclin) factors, contributing to endothelial dysfunction and an elevated cardiovascular risk profile. While previous studies have individually examined endothelial markers in PCOS, our study is the first to concurrently evaluate the diagnostic potential of both vasodilatory (NO, apelin) and vasoconstrictive (noradrenaline, prostacyclin) factors using ROC curve and logistic regression analyses. This integrative approach provides a more comprehensive understanding of vascular tone dysregulation in PCOS.

Polycystic ovary syndrome is a multifaceted endocrine condition that markedly affects vascular tone and endothelial function. Individuals with PCOS commonly experience impaired vascular responsiveness, potentially elevating their cardiovascular risk. This vascular dysfunction is driven by a combination of factors such as hormonal irregularities, insulin resistance, and elevated inflammatory markers [31,32]. NO is increasingly recognized for its role in modulating bioavailability and promoting endothelial vasodilation through enhanced production. The influence of NO on cellular processes is multifaceted and, at times paradoxical. It can exert cytotoxic effects while also serving as an antioxidant to protect against cellular damage. NO is implicated in the regulation of various physiological and pathological processes, including vascular tone, blood pressure, vascular remodeling, and inflammation [33,34]. Our findings indicate that serum NO levels in patients with PCOS are significantly elevated compared to the control group (Table 3). However, reported NO levels in PCOS patients exhibit considerable variability in the literature. While certain studies have documented decreased NO levels relative to controls [12,34,35], others have observed no significant differences between the groups [36,37]. Furthermore, Gao et al. reported that NO levels were higher in non-obese PCOS patients compared to the control group [38]. NO

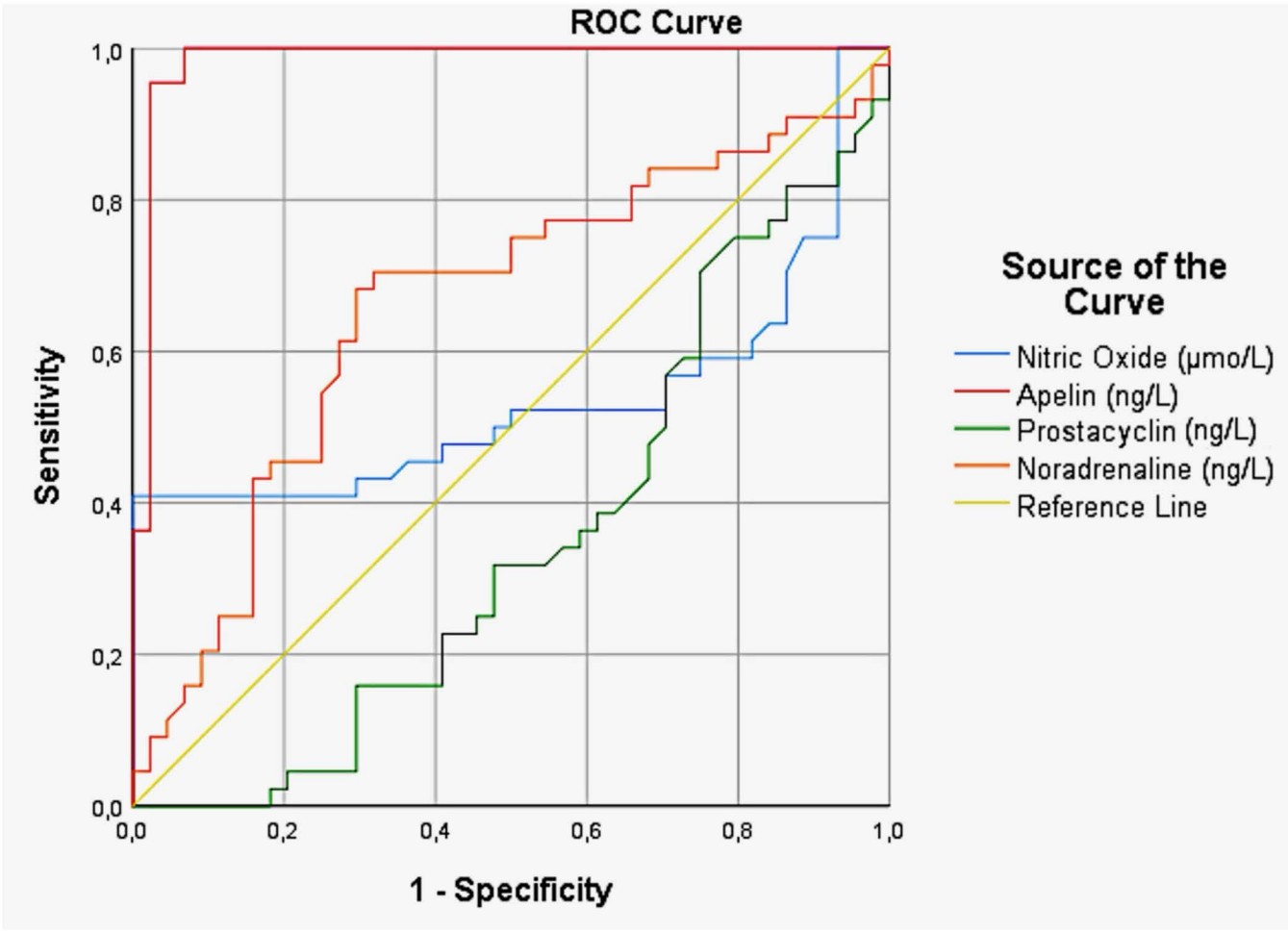

**Fig 1. ROC curve analysis of the utility of NO, apelin, noradrenaline, and prostacyclin to PCOS.**

(µmo/L) significantly predicted the difference between the patient and control groups (p = 0.012), with an odds ratio (OR) of 1.017 (95% CI: 1.004–1.031) (Table 4). In PCOS patients, the increase in NO levels can be attributed to the enhanced activity of endothelial NO synthase (eNOS), which is influenced by factors such as insulin resistance and hormonal imbalances. Elevated NO levels play a crucial role in regulating vascular tone, primarily promoting vascular relaxation. The ROC analysis for NO showed high sensitivity (97.7%) and specificity (93.2%), indicating that this parameter is a highly accurate marker for distinguishing between the patient and control groups. The AUC of 0.536 (p = 0.046) suggests a moderate predictive ability, making it a promising biomarker for assessing disease risk (Fig 1). However, this increase in NO may have a paradoxical effect, disrupt normal vascular responses and leading to imbalances in vascular tone. The mechanisms underlying this process are further complicated by the presence of inflammation and hormonal dysregulation, both of which can modulate NO production and contribute to endothelial dysfunction. These findings suggest that while NO's role in vasodilation is well-established, its overproduction in PCOS may contribute to vascular disturbances, warranting further investigation into its precise role in PCOS-related vascular pathology. In conclusion, the elevated NO levels in PCOS patients underscore its potential as a key player in vascular dysfunction and its role in regulating vascular tone. Despite some variability in the literature, NO remains a promising biomarker for assessing disease risk in PCOS.

Apelin is expressed in ovarian tissues, including granulosa cells and follicles, indicating its potential role in ovarian physiology. Elevated apelin levels are commonly observed in women with PCOS, especially those with obesity, and are associated with insulin resistance and metabolic disturbances. Studies have shown a correlation between serum apelin levels and BMI, as well as insulin sensitivity markers, indicating its potential as a biomarker for metabolic dysfunction in PCOS [24,39,40]. Some studies have reported conflicting results regarding the correlation between apelin levels and insulin resistance in PCOS patients. However, the relationship between apelin and PCOS is not entirely straightforward. For example, while elevated apelin levels have been associated with obesity and insulin resistance in some cohorts, other studies have found no significant differences in apelin levels between PCOS patients and healthy controls [41]. In a study by Altınkaya et al., BMI-matched comparisons revealed that women with PCOS had significantly lower apelin levels than controls, irrespective of whether they were overweight, obese, or of normal weight [42,43]. In this study, serum apelin levels were significantly higher in PCOS patients compared to healthy controls (Table 3). Apelin, through its interaction with APJ receptors in endothelial cells, plays a pivotal role in vasodilation and anti-inflammatory responses, making it an essential regulator of vascular tone. Apelin (ng/L) significantly predicted the group difference (p = 0.0001), with an OR of 1.044 (95% CI: 1.025–1.064) (Table 4). Apelin demonstrated an outstanding sensitivity of 98.7% and specificity of 95.5%, with an AUC of 0.983 (p = 0.001), which indicates that it is highly effective in differentiating between the patient and control groups (Fig 1). The AUC value, being very close to one, suggests that apelin is an excellent biomarker with a strong discriminatory power. The observed increase in apelin levels in PCOS may result from the excessive activation of this signaling pathway, potentially leading to impaired vascular responses and dysregulation of vascular tone. These findings suggest that while apelin is typically involved in maintaining vascular homeostasis, its overproduction in PCOS could contribute to vascular dysfunction and complications, such as endothelial dysfunction and altered blood pressure regulation. Given its strong discriminatory power, apelin may serve as a valuable biomarker for diagnosing PCOS and assessing its metabolic aspects. Further studies could explore the potential therapeutic implications of targeting the apelin/APJ signaling pathway in managing PCOS-related metabolic disturbances. Previous studies investigating NO and apelin levels in PCOS patients have reported variations in findings, with some suggesting increased levels, while others found no significant changes or even reductions. These inconsistencies may be attributed to differences in population characteristics, PCOS phenotypes, assay sensitivities, or study designs. In our study, we observed a concurrent elevation in both NO and apelin levels in the PCOS group compared to controls. Importantly, we also evaluated their diagnostic utility through ROC curve and logistic regression analysis. This dual approach not only strengthens the interpretability of these findings but also provides a more integrated understanding of their role in vascular tone regulation, thereby contributing to the resolution of previously reported disparities in the literature.

Noradrenaline, a neurotransmitter released by the sympathetic nervous system, plays a crucial role in regulating the stress response, vascular tone, and metabolic processes. Studies have shown that the sympathetic nervous system activation, evidenced by elevated noradrenaline levels, is associated with metabolic dysfunctions in PCOS, including insulin resistance and obesity [25,44]. Linares et al., investigating the role of the vagus nerve in modulating ovarian noradrenaline concentrations, found that unilateral or bilateral vagotomy had different effects on ovarian noradrenaline levels in rats with PCOS. Bilateral vagotomy was found to increase noradrenaline concentrations in the superior mesenteric ganglion of the celiac [45]. Musalı et al. have identified significantly higher levels of noradrenaline in the follicular fluid of PCOS patients compared to those with male infertility [46]. Robeva et al. demonstrated that plasma-free normetanephrine (NMN) and nerve growth factor (NGF) levels are elevated in individuals with PCOS, whereas renalase (RNL) levels are reduced compared to healthy controls [47]. According to our findings, serum testosterone, dehydroepiandrosterone sulfate (DHEA-S), and luteinizing hormone (LH) levels were found to be significantly increased in PCOS patients compared to healthy controls, whereas follicle-stimulating hormone (FSH) and sex hormone-binding globulin (SHBG) levels were decreased (Table 2). In our study, we observed significantly higher noradrenaline levels in PCOS patients compared to healthy controls (Table 3). This finding aligns with the notion that metabolic disturbances, which are commonly present

in PCOS, may contribute to excessive activation of the sympathetic nervous system. Noradrenaline showed statistical significance (p = 0.019), with an OR of 1.115 (95% CI: 1.018–1.221) (Table 4). The increased noradrenaline release, a key feature of sympathetic activation, can disrupt vascular responses, potentially impairing overall vascular tone regulation. Noradrenaline exhibited high sensitivity (93.2%) and specificity (95.5%), with an AUC of 0.659 (p = 0.010), which indicates its moderate to good ability in predicting the difference between the study and control groups (Fig 1). Nevertheless, it still contributes meaningfully to disease risk assessment. Additionally, PCOS is characterized by elevated androgen levels, which may further exacerbate sympathetic nervous system activation. The interplay between elevated androgens and sympathetic nervous system dysregulation highlights a complex mechanism that may contribute to the vascular and metabolic abnormalities seen in PCOS. Although noradrenaline's predictive value is moderate, its role in the pathophysiology of PCOS highlights its potential as a useful parameter for disease risk assessment. Further research is necessary to explore the full clinical relevance of noradrenaline in PCOS, particularly in conjunction with other biomarkers.

The concurrent elevation of NO and noradrenaline levels in PCOS may appear paradoxical, given their opposing roles in vascular tone regulation. However, this finding may reflect a compensatory response to ongoing vascular stress or dysfunction. Elevated noradrenaline, indicative of heightened sympathetic activity, could contribute to persistent vasoconstriction and endothelial stress. In response, the endothelium may upregulate NO production via inducible or endothelial NO synthase as a compensatory vasodilatory mechanism aimed at restoring vascular homeostasis. Alternatively, increased NO may be a consequence of oxidative stress and inflammation, common features in PCOS, which stimulate iNOS expression but do not necessarily lead to functional vasodilation. These complex and potentially maladaptive interactions suggest that the vascular system in PCOS operates in a dysregulated state, with both vasodilatory and vasoconstrictive pathways activated simultaneously but ineffectively. This imbalance may contribute to the observed endothelial dysfunction and increased cardiovascular risk in PCOS patients.

The role of prostacyclin in vascular biology is multifaceted, involving the inhibition of platelet aggregation and the modulation of vascular tone, both of which are crucial for maintaining hemodynamic stability [48,49]. Keller et al. investigated endothelial dysfunction in a rat model of PCOS induced by dihydrotestosterone (DHT), revealing impaired vasodilation and increased elastin content in resistance arteries. In DHT-treated rats, endothelial function was impaired, as evidenced by a significant reduction in vasodilation responses to acetylcholine (ACh). These findings suggest that prostanoids, particularly vasoconstrictor prostanoids, play a crucial role in DHT-induced endothelial dysfunction [50]. Elevated androgen levels may impair endothelial function, leading to a reduction in prostacyclin production [51]. The effects of androgens on endothelial cells can inhibit prostacyclin synthesis in PCOS. This condition can negatively affect vascular health and disrupt the regulation of vascular tone. Low-grade chronic inflammation, characterized by PCOS, may lead to a reduction in prostacyclin production [52]. Prostacyclin levels showed a significant difference between the patient and control groups, with lower levels observed in the patient group (Table 3). Prostacyclin (ng/L) was also a significant predictor (p = 0.007), with an OR of 0.658 (95% CI: 0.486–0.891) (Table 4). Prostacyclin, with a cut-off of 4.23, showed a sensitivity of 88.6% and specificity of 95.5%. However, the AUC of 0.351 (p = 0.016) was relatively low, suggesting that prostacyclin may have limited effectiveness in distinguishing between the patient and control groups on its own (Fig 1). Further research may be needed to assess whether prostacyclin's predictive value can be enhanced through combination with other parameters. Inflammatory mediators (e.g., TNF-α, IL-6) can inhibit prostacyclin synthesis, and the increase in oxidative stress also disrupts endothelial cell function. While the low AUC indicates that prostacyclin may not be suitable as a standalone diagnostic marker, its well-established role in endothelial function suggests it should not be overlooked. Rather than acting in isolation, prostacyclin may modulate or amplify the effects of other vasoactive molecules. Its reduced levels in PCOS, despite limited predictive power, may reflect a downstream effect of chronic inflammation or hormonal dysregulation. Integrating prostacyclin into multi-marker panels could improve its utility and provide more nuanced insights into PCOS-associated vascular impairment. Additionally, oxidative stress can reduce the effectiveness of prostacyclin, making the regulation of vascular tone more challenging.

While alterations in vascular biomarkers such as NO, apelin, noradrenaline, and prostacyclin have also been documented in metabolic conditions like obesity, metabolic syndrome, and type 2 diabetes mellitus, the pattern and extent of dysregulation observed in our PCOS cohort may represent a distinct pathophysiological signature. PCOS is characterized not only by insulin resistance and low-grade inflammation but also by a unique endocrine profile involving hyperandrogenism and ovulatory dysfunction. These hormonal imbalances may amplify or modulate vascular tone disturbances in ways not typically seen in other metabolic disorders. Although endothelial dysfunction is a common feature across insulin-resistant states, the concurrent elevation of NO, apelin, and noradrenaline alongside reduced prostacyclin levels -demonstrated through both group comparison and predictive modelling- highlights a potentially PCOS-specific vascular phenotype. It is also worth noting that there were no statistically significant differences in BMI, insulin, HOMA-IR, or HbA1c levels between the PCOS and control groups, indicating comparable anthropometric and glycemic profiles. Despite this similarity, significant alterations were observed in vascular markers. This finding strengthens the interpretation that the vascular changes in PCOS are not merely secondary to insulin resistance or obesity but may reflect PCOS-specific mechanisms such as hormonal imbalance and ovarian dysfunction.

In summary, this study expands current knowledge by integrating multiple endothelial markers and demonstrating their individual and combined diagnostic performance in PCOS. This approach enhances the understanding of endothelial dysfunction beyond conventional single-marker studies and opens the door for biomarker-guided clinical evaluations in PCOS. Given its outstanding diagnostic accuracy (AUC = 0.983), apelin stands out as a particularly promising biomarker. Its high sensitivity and specificity suggest a potential role not only in aiding the diagnosis of PCOS but also in complementing existing criteria such as the Rotterdam criteria. Incorporating apelin into a broader diagnostic panel may enhance diagnostic precision, particularly in patients with atypical or borderline presentations. Moreover, considering its association with vascular dysfunction, apelin could serve as a useful marker for stratifying cardiovascular risk in PCOS patients, paving the way for risk-based clinical management and personalized therapeutic approaches.

The limitations of this study include the relatively small sample sizes in both the patient and control groups, as well as its single-center design. Although the sample size was determined through a priori power analysis and fulfilled statistical requirements, future multi-center studies with larger and more diverse populations are needed to enhance the generalizability of these findings. Additionally, other parameters necessary for a more comprehensive evaluation of vascular tone were not assessed. Despite these limitations, our study provides valuable insights into the evaluation of vascular tone in patients with PCOS.

## Conclusion

The increased levels of vasodilators such as NO and apelin in PCOS patients contribute to enhanced vascular relaxation, while the elevated noradrenaline levels and reduced prostacyclin levels promote vasoconstriction. The simultaneous rise in apelin, NO, and noradrenaline may create an imbalance between abnormal vessel relaxation and constriction, leading to dysregulation of vascular tone and impaired vascular responses. Additionally, the core features of PCOS, including irregular cycles, polycystic ovary morphology, increased adiposity, insulin resistance, hormonal imbalances, and chronic inflammation, further complicate the production and effects of these molecules. These molecular alterations in PCOS can disrupt proper vascular function and potentially contribute to the development of cardiovascular issues. Further investigation into the interactions between these molecules is essential to gain a deeper understanding of the underlying mechanisms and to develop targeted therapeutic strategies.

The novel contribution of this study lies in its integrative evaluation of endothelial markers and the identification of apelin as a highly sensitive and specific biomarker for distinguishing PCOS patients from healthy individuals. This highlights a potential avenue for non-invasive diagnostic strategies in PCOS management.

## Supporting information

**S1 File. G*Power report.**
(PDF)

**S1 Data. Data file.**
(XLSX)

## Author contributions

**Conceptualization:** Tuğba Raika Kıran.

**Data curation:** Tuğba Raika Kıran, Feyza İnceoğlu.

**Formal analysis:** Tuğba Raika Kıran, Mehmet Erdem, Engin Yıldırım, Feyza İnceoğlu.

**Investigation:** Tuğba Raika Kıran, Mehmet Erdem, Engin Yıldırım.

**Methodology:** Tuğba Raika Kıran, Mehmet Erdem, Engin Yıldırım, Feyza İnceoğlu.

**Resources:** Tuğba Raika Kıran, Mehmet Erdem, Engin Yıldırım.

**Software:** Tuğba Raika Kıran, Feyza İnceoğlu.

**Supervision:** Tuğba Raika Kıran.

**Validation:** Mehmet Erdem.

**Visualization:** Tuğba Raika Kıran.

**Writing – original draft:** Tuğba Raika Kıran, Engin Yıldırım.

**Writing – review & editing:** Tuğba Raika Kıran, Mehmet Erdem, Feyza İnceoğlu.

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
