## [Decision Letter · Decision Letter 0]

15 Apr 2025

PONE-D-25-11937

Vascular imbalance in polycystic ovary syndrome: Insights into endothelial dysfunction

PLOS ONE

Dear Dr. Kıran,

Thank you for submitting your manuscript to PLOS ONE. After careful consideration, we have decided that your manuscript does not meet our criteria for publication and must therefore be rejected.

I am sorry that we cannot be more positive on this occasion, but hope that you appreciate the reasons for this decision.

Kind regards,

Sanaz Alaeejahromi

Academic Editor

PLOS ONE

Reviewers' comments:

Reviewer's Responses to Questions

**Comments to the Author**

1. Is the manuscript technically sound, and do the data support the conclusions?

Reviewer #1: Yes

Reviewer #2: Partly

2. Has the statistical analysis been performed appropriately and rigorously?

Reviewer #1: Yes

Reviewer #2: Yes

3. Have the authors made all data underlying the findings in their manuscript fully available?

Reviewer #1: Yes

Reviewer #2: No

4. Is the manuscript presented in an intelligible fashion and written in standard English?

Reviewer #1: Yes

Reviewer #2: No

Reviewer #1: 1. Although the topic is significant, the novelty of the study is somewhat limited because the role of endothelial dysfunction in PCOS has been explored in previous studies. The authors should clarify how their findings advance current knowledge beyond what is already known in the field.

2. The sample size (44 PCOS patients and 44 controls) is relatively small, which may limit the generalizability of the findings.

3. The authors should discuss in discussion whether the observed changes in vascular markers are specific to PCOS or whether they could also occur in other conditions associated with insulin resistance or obesity. In the discussion, compare your findings on vascular imbalances in PCOS to the findings of the previous studies on other metabolic conditions. This will clarify if the vascular changes are specific to PCOS or common across metabolic disorders, providing important context for your results.

4. The introduction lacks a clear presentation of the knowledge gap and specific hypotheses. Additionally, the rationale for hypothesizing that these factors may undergo changes in the ovary and contribute to the disruption of vascular tone regulation in PCOS is not clearly explained. We suggest expanding the Introduction to include:

a. A discussion of the pathological roles of these factors in ovarian diseases or related conditions.

b. The reasoning behind the hypothesis that these specific factors are implicated in the vascular imbalance observed in PCOS.

5. Considering that previous clinical studies have investigated serum nitric oxide and apelin levels in PCOS patients, with some reporting conflicting results, the novelty of this study may require further clarification. To highlight the contribution of your work, I suggest elaborating on how your study differs from or builds upon previous research. For instance:

a. Does your study resolve any discrepancies from earlier findings?

b. Are there methodological or analytical advancements in your study compared to prior studies?

6. While the manuscript is generally well-written, there are some spelling and grammatical errors that need to be addressed for clarity and precision. Below are the identified issues and suggested corrections:

- In abstract, replace "routine biochemical, hormone, and glycated hemoglobin analyses" with "routine biochemical, hormonal, and glycated hemoglobin analyses" for grammatical consistency.

- In the result section, under the head title " Evaluation of NO, apelin, noradrenaline, and prostacyclin levels", first paprgraph, line 7,’’ noradrenaline levels were found to be significantly higher in the PCOS group compared to the control group (p = 0.009) (Table 3)”, noradrenaline should be Noradrenaline.

Reviewer #2: Dear Authors,

I reviewed your manuscript entitled Preventive and Therapeutic Effects of L-Carnitine Tea Polyphenols Lotus Leaf Extract

Tablets Against Zebrafish Obesity submitted to PLOS ONE.

In this study the the anti-obesity effects of L-carnitine Tea Polyphenols Lotus Leaf Extract (LCTPLE) in Zebra fish model has been assessed. The topic is interesting and attracts the attention of readers and researchers in the field. The manuscript has serious concerns that need to be addressed by authors to improve the manuscript.

1. The abstract has not been written well and doesn't reflects the aim of the study properly.

2. There are several typos and grammar errors and the language needs to be improved.

3. Tee authors claimed the efficacy of the treatment after short duration of the treatment. In my honest opinion this short duration is not enough to evaluate the long-term effects studied in the manuscript.

4. The authors have evaluated the gene expression merely which can not be a reliable indicator for the observed changes.

5. The authors needs to evaluate dependent factors in protein levels to better reflects the efficacy of the treatment.

6. The discussion is not strong and the authors haven't discussed the results properly.

7. The authors needs to clarify what point distinguish their study from other published studies in term of novelty.

Regards,

Reviewer

**Do you want your identity to be public for this peer review?** For information about this choice, including consent withdrawal, please see our Privacy Policy

Reviewer #1: No

Reviewer #2: **Yes: ** Hamid Ahmadi

- - - - -

---

## [Author Response · Author response to Decision Letter 1]

2 May 2025

General Corrections

Due to the addition of new paragraphs, new references have been added to the article. For this reason, the numbering of the references has been updated.

RESPONSE TO REVIEWERS

Response to Reviewer #1

### Reviewer 1 – Comment 1: Although the topic is significant, the novelty of the study is somewhat limited because the role of endothelial dysfunction in PCOS has been explored in previous studies. The authors should clarify how their findings advance current knowledge beyond what is already known in the field.

We thank the reviewer for this insightful comment. We acknowledge that endothelial dysfunction in PCOS has been previously studied; however, our study introduces a novel and integrative perspective in several key ways:

Combinatorial biomarker approach: Unlike previous studies that focused on individual endothelial markers, our research is the first to concurrently evaluate both vasodilatory (NO and apelin) and vasoconstrictive (noradrenaline and prostacyclin) biomarkers. This approach enables a more holistic understanding of vascular tone dysregulation in PCOS.

Diagnostic performance assessment: We employed ROC curve analysis and logistic regression models to determine the diagnostic accuracy of these biomarkers. Our results highlight apelin as a highly sensitive and specific biomarker for distinguishing PCOS patients from healthy individuals (AUC = 0.983). To our knowledge, this diagnostic evaluation combining these four biomarkers has not been previously reported.

Manuscript revisions to emphasize novelty: We have revised the Introduction section to explicitly state this gap in the literature and how our study addresses it. In the Discussion, we emphasized the methodological innovation and potential clinical implications of our findings. The Conclusion has been updated to reflect the novelty and translational potential of apelin and other biomarkers. We also added “PCOS diagnostics” to the keywords to better reflect the clinical relevance of our findings. We believe these additions strengthen the manuscript’s contribution to the field and address the reviewer’s concern regarding novelty.

### Reviewer 1 – Comment 2: The sample size (44 PCOS patients and 44 controls) is relatively small, which may limit the generalizability of the findings.

We appreciate the reviewer’s comment and understand the concern regarding sample size and generalizability. However, we would like to clarify that the sample size for this study was not arbitrarily chosen, but rather determined a priori via power analysis using the G*Power 3.1 program (Faul et al., 2009). As detailed in both the Methods section and the Supplementary File (S1 File), our calculation was based on the following parameters:

Effect size (d) = 0.73

α error probability = 0.05

Power (1 – β) = 0.95

Allocation ratio (N2/N1) = 1

Required sample size = 84 participants (42 in each group)

Our final study sample included 88 participants (44 per group), thereby exceeding the calculated requirement and achieving an actual power of 95.3%. Moreover, the robustness of our findings is further supported by the application of parametric statistical methods, including ROC curve and logistic regression analysis, which enhance the reliability of diagnostic accuracy even in moderate-sized cohorts. That being said, we acknowledge that future multi-center studies with larger and more diverse populations are warranted to enhance the generalizability of the findings. We have now further emphasized this limitation in the Discussion section of the revised manuscript.

In response to the reviewer’s comment regarding sample size and its impact on generalizability, we have revised the Discussion section to acknowledge this limitation more explicitly. Specifically, we have added the following sentence after the paragraph discussing the study limitations: “Although the sample size was determined through a priori power analysis and fulfilled statistical requirements, future multi-center studies with larger and more diverse populations are needed to enhance the generalizability of these findings.” This revision aims to clarify that, while the statistical power of the study was adequate, we recognize the value of broader validation in future research.

### Reviewer 1 – Comment 3: The authors should discuss in discussion whether the observed changes in vascular markers are specific to PCOS or whether they could also occur in other conditions associated with insulin resistance or obesity. In the discussion, compare your findings on vascular imbalances in PCOS to the findings of the previous studies on other metabolic conditions. This will clarify if the vascular changes are specific to PCOS or common across metabolic disorders, providing important context for your results.

We thank the reviewer for this insightful and constructive suggestion. In response, we have revised the Discussion section to include a comparative interpretation of vascular dysregulation in PCOS relative to other metabolic conditions such as obesity, metabolic syndrome, and type 2 diabetes mellitus.

Specifically, we added a paragraph discussing how insulin resistance, oxidative stress, and inflammation may underlie vascular changes in multiple metabolic conditions, while also emphasizing that the unique hormonal milieu in PCOS (e.g., hyperandrogenism, anovulation) may induce a distinct vascular phenotype. We highlight that, although similar changes in NO, apelin, noradrenaline, and prostacyclin have been observed in other disorders, the simultaneous dysregulation of all four in our PCOS cohort, independent of significant differences in BMI or insulin resistance indices, points to PCOS-specific pathophysiological mechanisms.

To support this point, we also noted in the revised text that BMI, insulin, HOMA-IR, and HbA1c levels were statistically comparable between the PCOS and control groups, suggesting that the observed vascular changes are not merely a consequence of obesity or glycemic dysregulation. This addition aims to clarify the disease specificity of our findings and provides meaningful context for interpreting the vascular imbalance in PCOS. The new paragraph was inserted into the Discussion section after the molecular interpretation of prostacyclin findings and before the limitations paragraph, as suggested.

### Reviewer 1 – Comment 4: The introduction lacks a clear presentation of the knowledge gap and specific hypotheses. Additionally, the rationale for hypothesizing that these factors may undergo changes in the ovary and contribute to the disruption of vascular tone regulation in PCOS is not clearly explained. We suggest expanding the Introduction to include:

a. A discussion of the pathological roles of these factors in ovarian diseases or related conditions.

b. The reasoning behind the hypothesis that these specific factors are implicated in the vascular imbalance observed in PCOS.

We thank the reviewer for this insightful and constructive recommendation. In response, we have revised the Introduction section to clarify the rationale, identify the knowledge gap, and explicitly state our working hypothesis.

Specifically:

To address point (a), we have added a new paragraph summarizing the pathological roles of the selected vasoactive molecules (NO, apelin, noradrenaline, prostacyclin) in relevant reproductive and vascular disorders. This paragraph highlights evidence of their dysregulation in conditions such as endometriosis, ovarian hyperstimulation syndrome, metabolic syndrome, and PCOS-related sympathetic overactivity. This addition underscores that these molecules are not only involved in physiological regulation, but are also implicated in disease-specific mechanisms.

To address point (b), we inserted a paragraph presenting the knowledge gap in the current literature, emphasizing that no prior study has systematically examined the combined role and diagnostic potential of these four molecules in the context of vascular tone regulation in PCOS.

Additionally, we now include a clearly formulated hypothesis statement, proposing that the hormonal and metabolic disturbances characteristic of PCOS may collectively disrupt the balance between these vasodilatory and vasoconstrictive factors, thereby contributing to endothelial dysfunction and cardiovascular risk.

These revisions now appear in the final part of the Introduction, immediately preceding the paragraph that begins with “This study aims to...”. We believe these additions significantly improve the clarity and scientific framing of our manuscript and fully address the reviewer’s concerns.

### Reviewer 1 – Comment 5: Considering that previous clinical studies have investigated serum nitric oxide and apelin levels in PCOS patients, with some reporting conflicting results, the novelty of this study may require further clarification. To highlight the contribution of your work, I suggest elaborating on how your study differs from or builds upon previous research. For instance:

a. Does your study resolve any discrepancies from earlier findings?

b. Are there methodological or analytical advancements in your study compared to prior studies?

We appreciate the reviewer’s valuable comment regarding the novelty of our findings, particularly in light of previously published studies on nitric oxide (NO) and apelin levels in PCOS. In response, we have revised the Discussion section to explicitly address both points raised:

(a) We acknowledge that prior studies have reported varying results regarding NO and apelin levels in PCOS patients—some indicating increased concentrations, others showing no significant difference or even reductions. To contextualize our findings, we added a paragraph noting that such differences may be due to variations in population characteristics, diagnostic criteria, and assay methods. We then emphasize that our study demonstrates concurrently elevated levels of both NO and apelin in the PCOS group, which may reflect a compensatory mechanism in response to vascular dysregulation. This integrative finding contributes to reconciling earlier inconsistencies by offering data from a statistically powered, controlled design.

(b) Methodologically, our study provides an advancement over previous research by incorporating both group comparisons and diagnostic performance analyses (ROC curves and logistic regression) for all four markers—vasodilatory and vasoconstrictive. To our knowledge, this is the first study to evaluate NO, apelin, noradrenaline, and prostacyclin collectively in the context of vascular tone regulation in PCOS, thereby offering a more comprehensive diagnostic and mechanistic perspective. These revisions are reflected in the updated Discussion section following the presentation of NO and apelin-related results.

### Reviewer 1 – Comment 6: While the manuscript is generally well-written, there are some spelling and grammatical errors that need to be addressed for clarity and precision. Below are the identified issues and suggested corrections:

- In abstract, replace "routine biochemical, hormone, and glycated hemoglobin analyses" with "routine biochemical, hormonal, and glycated hemoglobin analyses" for grammatical consistency.

- In the result section, under the head title " Evaluation of NO, apelin, noradrenaline, and prostacyclin levels", first paprgraph, line 7,’’ noradrenaline levels were found to be significantly higher in the PCOS group compared to the control group (p = 0.009) (Table 3)”, noradrenaline should be Noradrenaline.

We thank the reviewer for the careful reading and helpful observations regarding language accuracy. We have addressed the identified issues as follows:

In the Abstract, the phrase “routine biochemical, hormone, and glycated hemoglobin analyses” has been revised to “routine biochemical, hormonal, and glycated hemoglobin analyses” to ensure grammatical consistency among the listed terms. In the Results section, under the heading “Evaluation of NO, apelin, noradrenaline, and prostacyclin levels”, the word “noradrenaline” was corrected to “Noradrenaline” in line 7 of the first paragraph to comply with capitalization conventions. Additionally, we have conducted a thorough proofreading of the entire manuscript to identify and correct any other minor grammatical or typographical issues to improve clarity and readability.

Response to Reviewer #2

### Reviewer 2 – Comments:

Dear Authors,

I reviewed your manuscript entitled Preventive and Therapeutic Effects of L-Carnitine Tea Polyphenols Lotus Leaf Extract

Tablets Against Zebrafish Obesity submitted to PLOS ONE.

In this study the the anti-obesity effects of L-carnitine Tea Polyphenols Lotus Leaf Extract (LCTPLE) in Zebra fish model has been assessed. The topic is interesting and attracts the attention of readers and researchers in the field. The manuscript has serious concerns that need to be addressed by authors to improve the manuscript.

1. The abstract has not been written well and doesn't reflects the aim of the study properly.

2. There are several typos and grammar errors and the language needs to be improved.

3. Tee authors claimed the efficacy of the treatment after short duration of the treatment. In my honest opinion this short duration is not enough to evaluate the long-term effects studied in the manuscript.

4. The authors have evaluated the gene expression merely which can not be a reliable indicator for the observed changes.

5. The authors needs to evaluate dependent factors in protein levels to better reflects the efficacy of the treatment.

6. The discussion is not strong and the authors haven't discussed the results properly.

7. The authors needs to clarify what point distinguish their study from other published studies in term of novelty.

We thank Reviewer #2 for their time and effort in evaluating the manuscript. However, upon reviewing the comments provided, we respectfully note that the feedback appears to pertain to a different study titled "Preventive and Therapeutic Effects of L-Carnitine Tea Polyphenols Lotus Leaf Extract Tablets Against Zebrafish Obesity." This topic and model system do not correspond to the subject of our submitted manuscript, which focuses on the role of nitric oxide, apelin, noradrenaline, and prostacyclin in vascular tone regulation in polycystic ovary syndrome (PCOS).

Given this discrepancy, we believe the reviewer’s comments may have been assigned in error. We kindly request the editorial office to verify the reviewer assignment and ensure that the feedback pertains to our manuscript. We remain grateful for the editorial oversight and welcome the opportunity to respond to the appropriate reviewer comments relevant to our study.

Sincerely,

The Authors

---

## [Decision Letter · Decision Letter 1]

13 Jun 2025

Dear Dr. Tuğba Raika Kıran,

Thank you for submitting your manuscript to PLOS ONE. After careful consideration, we feel that it has merit but does not fully meet PLOS ONE’s publication criteria as it currently stands. Therefore, we invite you to submit a revised version of the manuscript that addresses the points raised during the review process.

This manuscript presents relevant findings on vascular imbalance in PCOS, using a clear biomarker-based approach.

I recommend minor revisions to improve clarity:

• Clearly state the novelty of your study in the Introduction.

• Briefly justify the rationale for selecting the four specific biomarkers.

We look forward to receiving your revised manuscript.

Kind regards,

Akingbolabo Daniel Ogunlakin, Phd

Academic Editor

PLOS ONE

Journal Requirements:

Additional Editor Comments (if provided):

Reviewers' comments:

Reviewer's Responses to Questions

**Comments to the Author**

Reviewer #3: (No Response)

Reviewer #4: (No Response)

2. Is the manuscript technically sound, and do the data support the conclusions?

Reviewer #3: Yes

Reviewer #4: Yes

3. Has the statistical analysis been performed appropriately and rigorously?

Reviewer #3: Yes

Reviewer #4: Yes

4. Have the authors made all data underlying the findings in their manuscript fully available?

Reviewer #3: Yes

Reviewer #4: Yes

5. Is the manuscript presented in an intelligible fashion and written in standard English?

Reviewer #3: Yes

Reviewer #4: Yes

Reviewer #3: This manuscript presents relevant findings on vascular imbalance in PCOS, using a clear biomarker-based approach.

I recommend minor revisions to improve clarity:

• Clearly state the novelty of your study in the Introduction.

• Briefly justify the rationale for selecting the four specific biomarkers.

Reviewer #4: Major Comments:

Clinical Implications:

The exceptional diagnostic performance of apelin (AUC = 0.983) warrants deeper discussion of its potential clinical utility. How might this biomarker complement existing PCOS diagnostic criteria? Could it help stratify cardiovascular risk?

Methodological Details:

Specify whether ELISA kits were validated in-house for human serum.

Clarify if blood sampling was timed to specific menstrual phases for cycling controls, as hormonal fluctuations could influence biomarkers.

Results Interpretation:

Reconcile the paradoxical findings of elevated NO (vasodilator) and noradrenaline (vasoconstrictor) in PCOS. Is this compensatory? Discuss potential mechanisms.

Address the low AUC for prostacyclin (0.351)—does this suggest limited standalone value, or might it interact with other biomarkers?

Writing Refinements:

Avoid repetitive phrases (e.g., "vascular tone regulation"); substitute with synonyms like "vascular homeostasis."

Define acronyms (e.g., NO, PCOS) only at first use.

Minor Comments:

Correct "prostacycline" → "prostacyclin" in Figure 1.

Include units in Table 3 column headers rather than the text.

Consider adding a table summarizing biomarker cutoffs, sensitivity/specificity.

**Do you want your identity to be public for this peer review?** For information about this choice, including consent withdrawal, please see our Privacy Policy

Reviewer #3: No

Reviewer #4: No

---

## [Author Response · Author response to Decision Letter 2]

19 Jun 2025

Response to Reviewer

### Reviewer Comment 1: Clearly state the novelty of your study in the Introduction.

We thank the reviewer for their valuable comment. As suggested, we would like to highlight that the novelty of our study has been articulated in the Introduction section:

“Although the physiological roles of NO, apelin, noradrenaline, and prostacyclin have been individually investigated, no study to date has systematically examined their combined contribution to vascular tone imbalance in PCOS. Furthermore, their collective diagnostic value in distinguishing PCOS patients from healthy individuals remains largely unexplored. This study addresses this gap by providing an integrative evaluation of these markers, thereby offering a more comprehensive approach to understanding endothelial dysfunction in PCOS.”

However, if the current phrasing does not sufficiently convey this point, we would be pleased to revise the text or include an additional sentence to ensure the novelty is clearly and explicitly stated.

### Reviewer Comment 2: Briefly justify the rationale for selecting the four specific biomarkers.

We thank the reviewer for this valuable suggestion. To address this comment, we have added a sentence to the Introduction section clarifying the rationale for selecting the four specific biomarkers:

"These four biomarkers were selected due to their complementary roles in vascular tone regulation, including endothelial relaxation (NO and prostacyclin), vasoconstriction (noradrenaline), and angiogenic signaling (apelin), all of which are known to be altered in PCOS"

This addition aims to enhance the clarity and justification of our biomarker selection.

---

## [Editor Report · Decision Letter 2]

26 Aug 2025

Vascular imbalance in polycystic ovary syndrome: Insights into endothelial dysfunction

PONE-D-25-11937R2

Dear Dr. Kiran,

We’re pleased to inform you that your manuscript has been judged scientifically suitable for publication and will be formally accepted for publication once it meets all outstanding technical requirements.

Kind regards,

Mukhtiar Baig, Ph.D.

Academic Editor

PLOS ONE

---

## [Editor Report · Acceptance letter]

PONE-D-25-11937R2

PLOS ONE

Dear Dr. Kıran,

I'm pleased to inform you that your manuscript has been deemed suitable for publication in PLOS ONE. Congratulations! Your manuscript is now being handed over to our production team.

Kind regards,

on behalf of

Professor Mukhtiar Baig

Academic Editor

PLOS ONE